# *Cryptococcus gattii* Can Use the Cactus *Pilosocereus* spp. to Grow and Develop a Capsule and Produce Melanin In Vitro

Paola Ramos-Irizarry [1], Bárbara Sánchez [2] and Yaliz Loperena-Álvarez [3],*

1   Department of Molecular Microbiology and Immunology, Johns Hopkins University Bloomberg School of Public Health, Baltimore, MD 21205, USA
2   Department of Biology, University of Puerto Rico-Mayagüez Campus, Mayagüez 00681-9000, Puerto Rico; biss713@yahoo.com
3   Department of Biology, Pontifical Catholic University of Puerto Rico-Mayagüez Campus, Mayagüez 00681-9000, Puerto Rico
*   Correspondence: yaliz_loperena@pucpr.edu; Tel.: +1-787-406-3841

**Abstract:** *Cryptococcus gattii* is a pathogenic yeast, member of the *C. neoformans/gattii* complex. Previous work from our laboratory has established the presence of *C. gattii* on cacti lesions, providing proof that it can grow in a stressful environment. However, it is not known which part of the cactus the yeast uses for nutrients. The purpose of this research is to determine the ability of *C. gattii* to grow in different parts of the cactus to assess how the yeast adapts to grow in this unique environment. Cactus media were developed using the outer, inner, and whole cactus from *Pilosocereus* spp. *Cryptcoccus gattii* was grown on the different cactus media, along with potato dextrose agar as a control for 24 and 48 h at 30 °C. Compared to the control medium, yeast growth was reduced in all cactus media, while an increase in the capsule development of the yeast grown in the inner part and the whole-cactus media was observed. Interestingly, the yeast produces melanin when grown in the outer membrane medium, which was dependent on laccase, suggesting that the outer membrane may contain a precursor that stimulatates pigment production. To our knowledge, this is the first study addressing these key differences in the growth of *C. gattii* on different parts of the cactus.

**Keywords:** *Cryptococcus gattii*; *Pilosocereus* spp.; capsule; melanin; melanin ghost

## 1. Introduction

The *Cryptococcus neoformans/gattii* species complex are environmental fungi inhaled as yeast, which cause disease in humans, typically among populations with weakened immune systems. The encapsulated yeast *Cryptococcus gattii* is a respiratory pathogen that mainly affects immunocompetent individuals worldwide. *C. gattii* causes a disease known as cryptococcosis, which can manifest as a pulmonary infection or can disseminate to the brain, leading to fatal meningoencephalitis [1]. *C. gattii* shares many notable virulence factors with its sister species *C. neoformans*, including a polysaccharide capsule and the ability to produce melanin, both of which protect the yeast from the host's innate immune defenses [2–5]. The species is further subdivided into four molecular types: VGI, VGII, VGIII, and VGIV [6]. Of these, VGI and VGII infect immunocompetent individuals and VGIII and VGIV infect immunocompromised ones [7–9]. Cases are seen worldwide but are more prominent in sub-tropical regions, such as India, South America, Australia, and Papua New Guinea, and HIV-prevalent regions, such as Sub-Sharan Africa [10,11]

Recently, there has been an increase in reported cases of *C. neoformans* and *C. gattii* in more temperate places, such as the United States, Europe, and Canada [12,13]. To date, *C. gattii* has been found in six continents and approximately 41 countries [11]. Previous reports indicated that the yeast is mainly found in tropical and sub-tropical climates in the detritus of eucalyptus and almond trees. The dispersal of *C. gattii* outside Australia

in recent years has suggested that eucalyptus trees, though exported to tropical and sub-tropical places, are not the main source of the yeast [14,15]. Studies in the Brazilian Amazon rainforest, a place unaffected by the exportation of eucalyptus trees, found yeast in the detritus of endemic trees in the region, suggesting that either detritus or organic soil is likely the main ecological niche of the yeast [16]. Other tropical and sub-tropical regions where *C. gattii* has been identified are India, Thailand, Peru, Cuba, Mexico, Northern Australia, among other countries [11,17]. In these regions, the main ecological niches have been detritus from trees, including almond tree, eucalyptus, Indian lilac, river red gum, Spanish cherry, among others. Infections have also varied in hosts, with *C. gattii* not only infecting humans in these regions, but also animals, such as dolphins and cats [11,16,18].

Aside from the ecological source, the climate plays an important role in the growth of yeast. In 2002, a *C. gattii* outbreak was reported in the Vancouver region, where an environmental survey was conducted to identify the ecological source of the yeast. This outbreak affected 59 individuals [7,19–21]. The dissemination of the yeast in Vancouver, in comparison to Brazil, shows a striking change in its distribution. Studies in the area found that even though eucalyptus trees were present, the yeast was found in the detritus of other endemic trees in the Canadian forest, such as the Canadian island pine tree [18,22]. A similar temperate climate is shared by Northern Europe, as *C. gattii* has been found in the Netherlands [23]. In contrast, the yeast has been isolated in the Southern European regions where the climate is strikingly different, characterized by dry, hot summers and mild winters. The presence of *C. gattii* in this area was another interesting change in environmental distribution [24]. This area is abundant in olive and eucalyptus trees; the latter is a known reservoir for this pathogen and the former is new. Although trees and soil have been established as the main reservoirs of this pathogen, the yeast ecological diversity begs the question as to how limiting it is to exclusively sample these niches [12,13].

Our lack of understanding of the ecological niche of this pathogen has made us overlook the ability of *C. gattii* to adapt to a wide array of niches, widening its distribution and epidemiological reach. To further expand on this point, the ecological diversity of this pathogen was again brought into question when it was identified in Puerto Rico. In 2010, it was found for the first time in cacti from the Guánica Dry Forest, a niche previously thought to be a hostile environment for the yeast. The cactus was identified as *Pilosocereus (Cephalocereus) royenii* [25]. Although Puerto Rico's climate is tropical with both humid and dry forests, cacti constitute a new ecological niche that has not been explored. Furthermore, the southern part of the island has similarities in climate with other regions where *C. gattii* has been found, such as the Mediterranean Basin.

The discovery of the yeast in such a hostile, dry environment poses many questions about the capacity of the yeast to adapt to different ecological niches. Additionally, the properties that the cactus possesses that facilitate the growth of *C. gattii* are unknown. Currently, to the best of our knowledge, there have been no studies that show how the yeast behaves when it is grown on cacti, representing a gap in our knowledge. The purpose of this research is to determine which part of the cactus allows *C. gattii* to grow and to examine the behavior of the cells in terms of morphology, growth capability, and melanin production. These results will allow us to better understand how the yeast adapts to new environments using the cacti as a transitory niche.

## 2. Materials and Methods

### 2.1. Cell Culture

*Cryptococcus gattii* strain R265 (VGIIa) Vancouver Island clinical isolate [7], kindly provided by the Dr. Arturo Casadevall laboratory, was grown in yeast extract peptone dextrose broth (YPD) (BD Difco$^{TM}$ cat. no. 90003-284) from a $-80\,^{\circ}$C culture for 24 h at $30\,^{\circ}$C in a Digital Cel-Gro Tissue Culture Single Drum Rotator culture wheel (Thermo Fisher$^{TM}$ cat. no.88882016) at 60 rpm.

### 2.2. Cactus Media

The cactus *Pilosocereus* spp. was used to develop the cactus media (Planetdesert.com and succulentmarket.com accessed on 1 July 2021; online stores in Southern California). The cactus was divided into two different parts in order to prepared the cactus media. The outer part medium (OPM), composed mainly of the skin, cortex, and spines; the inner part medium (IPM), composed of the vascular tissue and the pith. Additionally, a whole-cactus medium (WCM) was prepared using all parts of the cactus. To prepare the cactus extract, the cactus was washed with MiliQ water and cut into pieces and 10 g of each part was ground with 100mL of distilled water using a blender. The mixture was filtered using a cheesecloth, and the resulting filtrate was autoclaved 121 °C 121 pi for 30 min. This extract was used as cactus broth. To prepare the cactus agar, 3 g of Bacto agar was added to 100 mL of the extract, followed by autoclave sterilization. Potato dextrose broth and agar (PDB, PDA) (BD DifcoTM cat no. 90003-494) were selected as control media for this experiments because it is rich in carbohydrate which resemblance the cactus composition.Additionally, PDA/PDB do not have all the nutrients that the yeast needs for its optimal growth [26,27].

### 2.3. Growth Curve

*C. gattii* strain R265 was grown in YPD broth from a $-80$ °C culture for 24 h at 30 °C in constant shaking. A cell count was performed, and $5 \times 10^5$ cells were inoculated into 2 mL of the different cactus media and into PDB control in a 12-well plate. The plate was incubated in a SpectraMax M5 for 96 h with constant shaking at 26–28 °C. OD-600 readings were collected every 2 h. Statistical analysis was performed using a two-way ANOVA with Graph pad Prism 9 software, Dotmatics, Boston, MA, USA.

### 2.4. Microscopy and Capsule Measurements

*C. gattii* R265 was grown on YPD broth for 24h in a culture wheel at 60 rpm. After the incubation period, $5 \times 10^5$ cells were inoculated in tubes with the OPM, IPM, WCM and into PDB as a control. Negative staining with India ink was performed using a 1:1 ratio of ink to the cell cultures at 24 and 48 h time points by mixing in a 1.5 mL volume and then pipetting 5 μL onto a microscope slide. Imaging was performed using an Olympus AX70 microscope and a QImaging Retiga 1300 digital camera with the QCapture Suite V2.46 software. Capsule measurements were performed using a Quantitative Capture Analysis program developed by Dragotakes and Casadevall, 2018. For each medium condition, one hundred cells were measured.

### 2.5. Inoculation of Cryptococcus gattii into Pilosocereus spp.

*Pilosocereus* spp. was obtained as described above. After disinfecting with a 70% ethanol solution, small incisions were performed crossing from the outer part into the inner part. The incision was inoculated with $2.5 \times 10^5$ cells of *C. gattii* for 12 days at room temperature. For quality control purposes, after Day 12 of incubation, swabs from each cactus were collected and grown in YPD and stained by India ink to confirm that the only microbe growing was the yeast.

### 2.6. Melanin Ghosts

For these purposes, the protocol developed by Wang [28] and modified by Chatterjee [29] was used. Briefly, *C. gattii* strain R265 was grown in 5 mL of YPD Broth for 24 h at 30 °C in a culture wheel at 60 rpm. After 24 h, *C. gattii* R265 was inoculated into the different cactus media in a 1:10 ratio. Minimal Media with L-dopa (15 mM D-glucose, 10 mM MgSO$_4$•7 H$_2$O, 20.3 mM KH2PO$_4$, 3 mM Glycine, 10 mg/mL Thiamine, pH 5.5) was used as the control. The cultures were incubated for 10 days at 30 °C with constant shaking. On Day 10, the culture was centrifuged at 4000 rpm for 10 min to pellet cells. Media were discarded and cells were washed with 1X PBS (pH 7.4) twice. Cells were then subjected to an acid hydrolysis at 80 °C for an hour to remove the cell walls, proteins, and lipids from the melanin layer. Next, the mixture was left to cool, and cells were centrifuged at 4000

rpm for 10 min. The supernatant was discarded and the remaining cells were washed with 1X PBS (pH 7.4) 3 times. Cells were resuspended in 1 mL of 1X PBS (pH 7.4) and stored at room temperature. Cells were then mounted on slides and imaging was performed using an Olympus AX70 microscope and a QImaging Retiga 1300 digital camera with the Qcapture Suite V2.46 software to determine the presence of melanin ghosts.

## 3. Results

### 3.1. Growth Differences of C. gattii in Cactus Media/Broth

To determine if the cactus possesses the necessary nutrients for the growth and proliferation for *C. gattii*, strain R265 was grown for 96 h in the three different cactus media (OPM, IPM, WCM) and compared to the PDB control medium. Between the three cactus media, the one that showed the fastest growth rate (up to 40 h) was IPM. By 48 h, all three cactus media curves leveled out, and no significant differences between the curves were observed (Figure 1). However, a significant reduction in the growth rate of *C. gattii* in the three cactus media was observed when compared to the control, as early as 24 h and remained the same at 80 h with IPM vs. PDB ($p = 0.0048$), OPM vs. PDB and WCM vs. PDB ($p = 0.005$) (Figure 1). These results indicate that the cactus-based media support the growth and proliferation of *C. gattii*, but not as efficiently as PDB, suggesting that possibly play a role as a transitory ecological niche for the yeast.

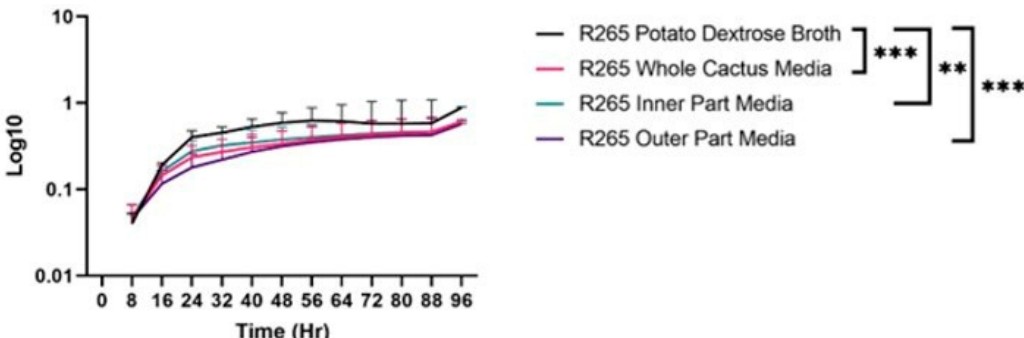

**Figure 1.** *Cryptococcus gattii* strain R265 can grow in different cactus media. Growth curve of *Cryptococcus gattii* strain R265 in outer part media (OPM), whole-cactus media (WCM), and inner part media (IPM) of the cactus *Pilosocereus* sp. and compared to potato dextrose broth (PDB) (control) after 96 h at 30 °C. Graph shows an XY with 3 repeated measurements aligned for each point. *X*-axis is time in hours; *Y*-axis is O.D. 600 Log10. Significant differences in O.D 600 measurements was found in PDB vs. OPM ($p < 0.0001$, Tukey's multiple comparisons); PDB vs. WCB ($p < 0.005$, Tukey's multiple comparisons); PDB vs. IPM ($p < 0.0048$, Tukey's multiple comparisons); (** = 0.0048, *** = 0.005).

### 3.2. Analysis of Cell Body and Capsule Radius of C. gattii in Cactus Media

To determine differences in cell body and capsule radius, *C. gattii* was grown in the three cactus media, negatively stained with India ink, and measured at 24 and 48 h (Figure 2). India ink staining of cells grown in IPM and WCB show significantly larger capsules compared to the cells grown in OPM and the control (Figure 2A). When a the capsule radius was analyzed, cells grown in IPM had a significantly larger capsule compared to the cells grown in the OPM and WCB medias and control ($p > 0.0001$, Dunnett's T3 multiple comparisons) (Figure 2B,C). Lastly, it was determined that *C. gattii* grown in IPM, OPM, and WCB had a significantly smaller cell body radius when compared to cells grown in control media at 24 h ($p > 0.0001$, Dunnett's T3 multiple comparisons) and 48 h ($p > 0.0001$, Dunnett's T3 multiple comparisons) (Figure 2B,C). As hypothesized, these results suggest that the cactus does not provide optimal nutritional sources for *C. gattii*. This is shown in the enlargement of the capsule when grown in IPM, mainly the result of stress conditions [4] and the significantly smaller cell body in all cactus media. It also suggests that the composition of IPM is different from OPM.

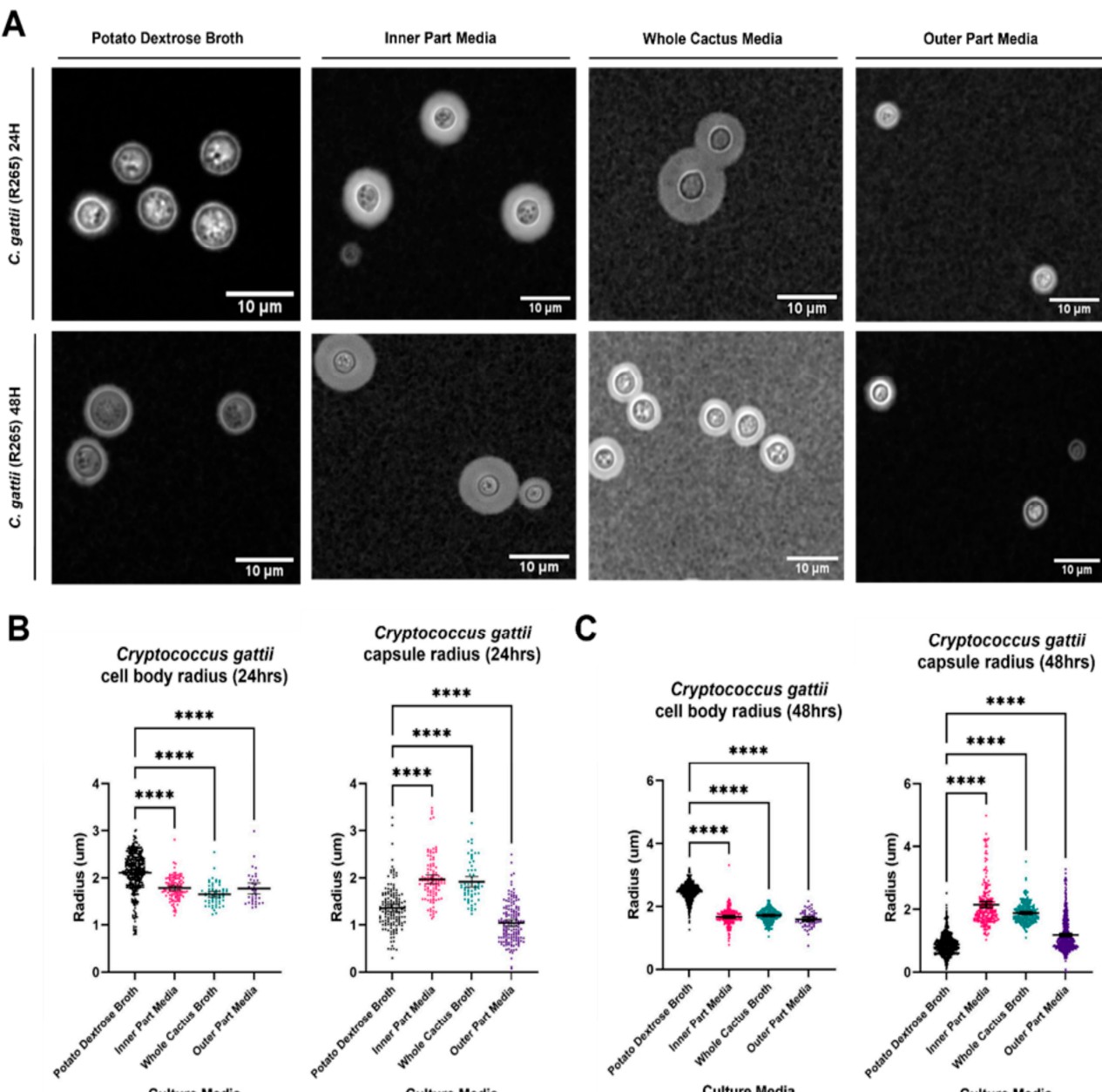

**Figure 2.** C. gattii growing in the different cactus media at 24 h (**A**) and 48 h (**B**). Differences in cell body and capsule size of *Cryptococcus gattii* strain R265 grown in cactus media. India ink staining of *C. gattii* R265 growth in different media conditions: outer part media (OPM), whole-cactus media (WCM), and inner part media (IPM) compared with potato dextrose broth (PDB) (control) after 24 or 48 H at 30 °C. Cell body radius and capsule radius measurements of *C. gattii* R265 growth in different media after 24 H obtained by QCA analysis. A significant decrease in cell body radius was observed in *C. gattii* growth in all 3 cactus media conditions when compared to control. Whole-cactus media being the smallest (**** $p < 0.0001$). A significant increase in capsule radius was found in *C. gattii growth* in all inner part and whole-cactus media when compared to the PDB control ($p < 0.0001$). Outer part media showed the smallest capsules, followed by the PDB control. (**C**) Cell body radius and capsule radius measurements of *C. gattii* R265 growth in different media conditions after 48 H obtained by QCA analysis. A significant decrease in cell body radius was found in *C. gattii* growth in all 3 cactus medias conditions when compared to PDB control with cells grown in whole-cactus medium being the smallest (**** $p < 0.0001$). A significant size increase in capsule radius was found in *C. gattii* growth in all inner part and whole-cactus Media when compared to the PDB control

($p < 0.0001$). Outer part media showed the smallest capsules, followed by the PDB control. Inner part media showed the biggest capsules. Graphs show an OPMtter plot with mean + 95% CI. Significance was measured by 2-way ANOVA and Tukey's multiple comparisons. Images were taken by an Olympus AX70 microscope and a Qimaging Retiga 1300 digital camera with the Qcapture Suite V2.46 software. Image analysis was performed by Fiji: Image J.

### 3.3. Pigment Produced by C. gattii When Grown in Outer Part Broth Is Laccase-Dependent

To observe if there were any differences between the colony morphologies of *C. gattii*, cactus media agar was developed to observe the growth of the colonies at 30 °C. After 3 days of incubation, colonies were be observed on the control plates, but not on the IPM or OPM plates (Figure 3A). Faint colonies were observed on Day 5 post inoculation for both IPM and OPM. An unexpected finding was the existence of a dark pigment in the colonies grown in OPM, which was absent from colonies grown on IPM and control plates. The dark pigment observed in the OPM plates also seemed to be cell-density-dependent, as it appears to reduce with the decreasing number of cells. These results suggest that OPM has precursors used for pigment production by the yeast. One such pigment is melanin, which is widely known to be produced by the yeast under stress conditions as a defense mechanism.

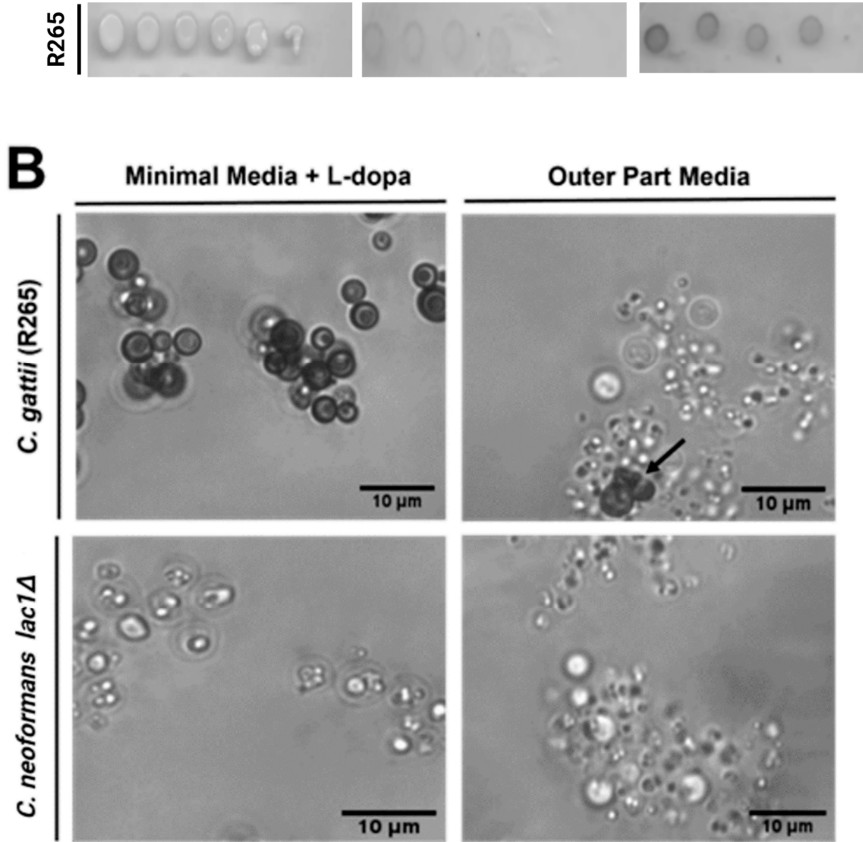

**Figure 3.** *C. gattii* strain R265 grown in cactus outer part media produces a dark pigment. (**A**) Plate images of *C. gattii* R265 grown in potato dextrose broth (PDB), our control, and 2 cactus media conditions, inner part (IPA) and outer part (OPA), after 37 °C. Serial dilutions were plated $10^8$–$10^1$. (**B**) Images melanin ghosts: top panel is *C. gattii* strain R265 grown in minimal media + L-dopa or outer part media (OPM); bottom panel is *C. neoformans* Δ*lac1* grown in minimal media + L-dopa or outer part media. Images taken by an Olympus AX70 microscope and a Qimaging Retiga 1300 digital camera with the Qcapture Suite V2.46 software and analyzed by Fiji: Image J.

Our next approach was to explore if the pigment observed in the OPM plates is melanin. For these experiments, *C. gattii* R265 and a *C. neoformans lac1Δ*, which has the laccase gen silenced, were used to produce melanin ghosts (Figure 3B). Melanin production is dependent on laccase since this enzyme helps melanin travel to the fungal cell wall [30]. For the *C. gattii* cells grown in melanin-inducing media, we observed the expected melanin ghosts (Figure 3B, top left image), and for *C. neoformans lac1Δ* in the same media, we observed cell debris, showing that there indeed was no melanin production (Figure 3B, bottom left image). For the *C. gattii* cells grown in the OPM, we did observe a small amount of melanin ghosts, not as clear as the one observed in melanin-inducing media; this is supported by the pigment that the melanin ghost sample still had after boiling in acid (Figure 3B, top right image). Yet, the dark color of the sample and the number of melanin ghosts identified by microscopy were significantly lower than our *C. gattii* melanin media sample. This suggests that the OPM contain a precursor to the stimulation of pigment production in *C. gattii*, but it may only exist in small amounts.

### 3.4. Melanin Is Produce in Cactus When C. gattii Is Directly Inoculated

To determine if the melanin precursor was indeed in low concentrations or that it was diluted when the media was develop, *C. gattii* was inoculated directly onto the cactus. To rule out the possibility of this dark pigment rotting, a *C. neoformans lac1Δ* was used for comparison along with an un-inoculated piece of cactus, both as control. After 12 days of incubation, the piece of cactus with C. gattii R265 had a dark brown pigment, while the *C. neoformans lac1Δ* did not (Figure 4A, top left, and top middle image). Our un-inoculated piece of cactus also remained without pigment (Figure 4A, top right image). This suggests that the possible precursor may exist in small amounts in the cactus and it was diluted during the preparation of the outer part broth. It also supports the idea that the color change that was observed is not due to the rotting of plant material, but the growth of the yeast itself.

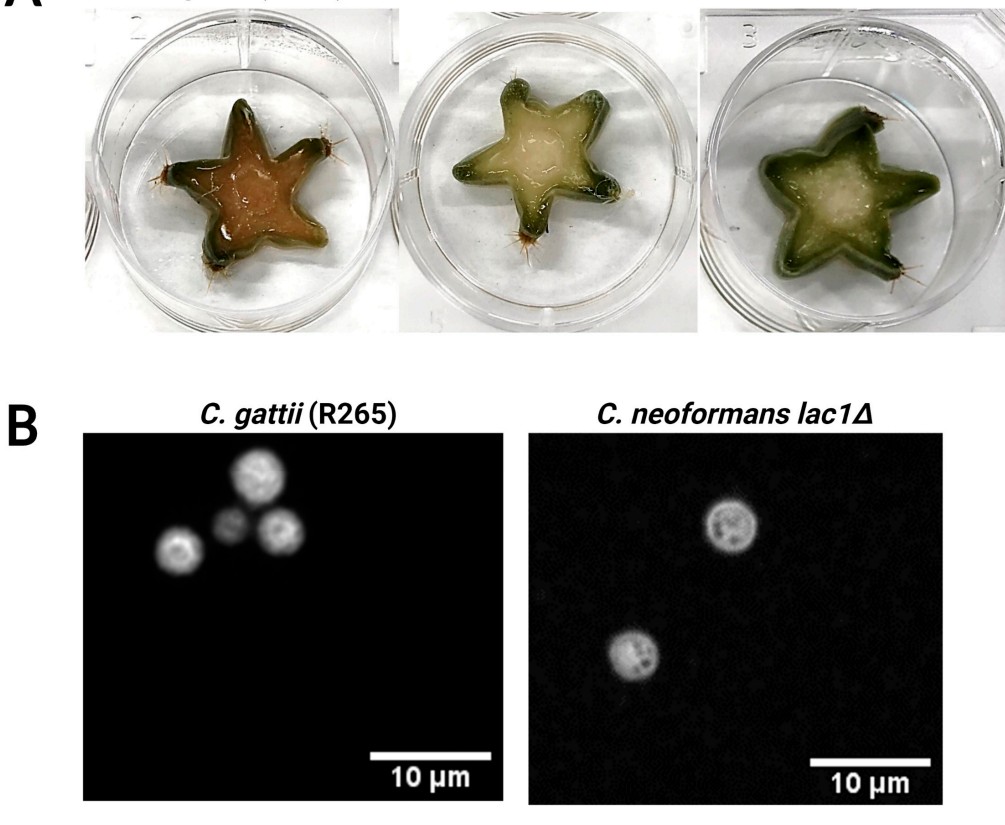

**Figure 4.** *Cryptococcus gattii* and *neoformans* directly grown on *Pilosocereus* spp. pieces to identify if pigment production is laccase-dependent. To determine if pigment production is laccase-dependent

and that pigment observed is melanin and not rotting plant material, *C. gattii* and *C. neoformans* were directly grown on cactus pieces. (**A**) *Pilosocereus* spp. pieces inoculated with $5 \times 10^5$ of *C. gattii* strain R265 and our positive laccase–control *C. neoformans* Δ*lac1* for 10 days. (**B**) Cells were collected by swab, and India ink was performed to verify that the only microbe growing on the cactus pieces was the yeast. Images were taken by an Olympus AX70 microscope and a QImaging Retiga 1300 digital camera with the QCapture Suite V2.46 software, and analyzed by Fiji: Image J.

## 4. Discussion

The results of this research are the first ones that address the capability of the yeast, specifically *C. gatttii* VGII, to use cacti as sources of growth and reproduction, which can explain how it adapts to survive in this environment. Since *C. gattii* VGII can grow in the cacti, we propose that the cacti can be a transitory niche for the yeast. From the research conducted by Loperena-Alvarez et al., 2010 [25], the cacti from which *C. gattii* was initially isolated were *Melocactus intortus* and *Pilosocereus royenii*. Since these species are protected by the Department of Natural Resources of Puerto Rico, a related species, *Pilosocereus* spp., was used in this research. This cactus is known as a columnar cactus, which is characterized by rigid arm-like protrusions and a succulent cortex, which help them gather water and nutrients, while protecting the inner structures from the arid desert environment [31]. These internal structures (inner part) are known as the vascular tissue that resides in the inner portion surrounded and protected by the cortex and skin (outer part). The molecular composition of these parts has not been fully studied, but the consensus is that the vascular tissue is made up of the xylem and phloem, structures plants use for storing and trafficking water and nutrients [31]. The control media, PDA/PDB, used in the experiments aimed to simulate the nutrient conditions in the cactus, mainly carbohydrates. Our results indicated that *C. gattii* can grow in all three different parts of the cactus. This result was unexpected since it was thought that the yeast may grow only inside the cactus because it represents the area where more hydration is available. In the work performed by Loperena-Alvarez et al., 2010 [25], the samples were collected from cactus lesions and not the cortex. In this research, to further study the capability of *C. gattii* to use the cactus parts for development, a growth curve was performed (Figure 1). The growth on the cacti limits the full development of the cell in size and capsule development (Figure 2). When compared to control, the growth observed in all three of the cactus media was suboptimal, supporting our hypothesis that the yeast can use the cacti as transitory niches.

Analysis of cell morphology focusing on the cell body and capsule radius was performed. From the three cactus media, the one that had the cells with the biggest cell bodies was the inner part media, followed by the whole-cactus media, and outer part media were last (Figure 2). In contrast, IPM- and WCM-grown cells had larger capsules when compared to OPM and control alike. Further, cell counts and the microscopy of *C. gattii* samples grown in IPM showed the most cell proliferation and budding cells of the three cactus media. According to the reports by Loperena-Álvarez et al. 2010 [25], it was expected that the IPM allows for better development of the yeast since this part is protected from UV light the most, is the most dissected, and has the most humidity. As previously stated, this section is known to store nutrients, specifically starches such as glucose and fructose, which are produced in the cortex of the cactus and then trafficked to the inner part. This information alone does not explain the capsule enlargement observed in these media; however, capsule-inducing conditions for *Cryptococcus* are known to be related to slower growing environments [32]. Most capsule-inducing media try to mimic the mammalian host environment since capsule enlargement is one of the yeast's defense mechanisms during infection. Other media achieve this enlargement by diluting rich media (Sabouraud's agar) and modifying to a neutral pH or by chemically defining it and creating it from scratch (minimal media) [32,33]. Our IPM most likely contains a sugar such as glucose and has a pH of 4.9, conditions most like the ones found in minimal media. However, we do not know any other components the cacti may provide, which begs the

question of whether low-sugar conditions and a slightly acidic pH are enough to induce capsule formation in cacti.

OPM-grown cells exhibit small cell bodies and capsules, which begs the question of whether there is a different in this media compared to IPM. The cortex of the cactus is a place where sugar production occurs due to the presence of the chlorenchyma, but they are rapidly trafficked to the vascular tissue inside the cactus. This suggests that the sugar content is significantly less. This, paired with a pH of 4.8 and dopamine analogs, may not be enough to induce an enlargement of the capsules or cell bodies. Additionally, studies suggest that capsule formation is linked to mitochondrial activity and media/environments that disrupt this yield smaller capsules [34]. Although it is unknown if *Pilosocereus* spp. contains any substances in the cortex that inhibit mitochondrial activity, other cacti species within this genus are known to have trypsin inhibitors, which could affect mitochondrial activity [35]. The presence of vesicles, such as deformations in the cells grown in all three cactus medias and PDB samples, were observed after performing India ink staining (Figure 2). It is hypothesized that these deformations could be due to extracellular vesicles that carry capsule or cell wall components, a matter that deserve to be further investigated.

An unexpected finding in the cactus media samples was the production of pigment, by the yeast. The most common pigment produced by *C. gattii* is melanin, which the yeast produces when precursors such as catecholamines are available. During infection, this pigment protects the yeast against oxidative bursts, cytokine attacks, and phagocytosis. In the environment, it is produced as a protection mechanism against sun radiation, a desiccation, hence this is a vital component for yeast survival [36–39]. In vitro, certain media have specific ingredients to recreate these conditions such as using staib media in nutritive media or L-dopa in minimal media; however, this ingredient was not added in the cactus media used in this investigation. This poses the question of what component of the cactus media is being utilized by the yeast to produce pigment and whether this pigment is melanin. To study this, we grew *C. gattii* R265 in outer part media alongside *C. neoformans lac1Δ*. Although the *C. neoformans* laccase mutant is another species, the results are comparable since melanin production uses the same. In the case of *C. gattii*, since it has been found to be more virulent, the expression of the melanin-associated genes is greater, thus producing more melanin [30,40,41]. Another difference that is relevant but does not affect the melanin synthesis pathways is the way melanin is deposited in the cell wall due to the amount of chitosan available [39]. In this research, our results demonstrated that *C. gattii* produces a pigment in a laccase-dependent manner, leaving the possibility that this pigment may be melanin. To verify if the pigment being produced was melanin, the production of melanin ghosts was induced. It was observed that *C. gattii* R265 grown in OPM yielded a small amount of melanin ghosts in comparison to positive control grown in MML-dopa. *C. neoformans lac1Δ* did not yield any melanin ghost in either media condition. Research performed by Tâniada Silveira Agostini-Costa (2020) [42] found that in the skin of *Pilosocereus* sp. dopamine analogs can be found, supporting our findings. The low pigment production and small number of ghosts produced in the OMP by *C. gattii* may be indicative of this precursor not being available in the right quantities, confirmation of the fact that the trafficking is being reduced for the yeast to produce melanin. This question was addressed by the direct inoculation of *C. gattii* onto the cactus. As observed in Figure 4, the pigment production was substantially more than the one produced in the media, which supports the fact of a possible dilution of the melanin precursors in the cacti. Regardless, the ability of *C. gattii* to be melanized in this substrate may provide protection from UV light and desiccation when growing in the cactus in the appropriate environment, providing another adaptation for this yeast to survive in this novel niche.

## 5. Conclusions

In conclusion, to the best of our knowledge, this research represents the first study addressing the cactus as a transitory niche of *C. gattii*. This is supported by the fact that the yeast could grow in all the parts of the cactus but presented the most cell proliferation

when grown using the inner part media. Additionally, cells grown using the inner part media developed the biggest capsules, suggesting that this part has the most nutrients supporting capsule development and the growth of *C. gattii*, while still being low in nutrients. Furthermore, the induction of big capsules and melanization provide us with some insight into the ability of *C. gattii* to use known virulence factors as possible adaptation mechanisms to survive in a different environment. In terms of how this benefits the yeast inside a human host, studies have shown that cryptococcal cells with bigger capsules result in worse disease outcomes for the host. The way *C. gattii* managed to adapt to this novel niche and use its virulence factors for survival suggests that this environment could be a breeding ground for more virulent strains. However, more research is needed to find the properties and composition of these media and see if ecological isolates from this environment elicit such results.

**Author Contributions:** All the experiments described, and the writing of this manuscript were performed by P.R.-I. B.S. supported the experimental part by growing the yeast for these experiments in UPRM Biology facilities. Y.L.-Á. developed the project, guided Paola in the experiments, proofread the article and wrote part of the discussion of this manuscript. All authors have read and agreed to the published version of the manuscript.

**Funding:** This research received no external funding.

**Data Availability Statement:** Data can be obtained by writing to Yaliz Loperena-Álvarez.

**Acknowledgments:** We would like to thank the Department of Biology of the University of Puerto Rico-Mayagüez campus for allowing us to use their microbiology facilities to perform the first experiments of this project. Additionally, we want to thank Alejandro Ruiz-Acevedo for his suggestions when developing the research plan to test our hypothesis. Follow up experiments and data utilized in this manuscript were obtained in Arturo Casadevall's Lab at Johns Hopkins Bloomberg School of Public Health. We would like to thank him for giving us access to his laboratory space and materials to complete this story. We would also like to thank Alex Ramirez, Rossana Baker, and Daniel Smith for their help in the experiments through which we made the cactus media and melanin ghost development. Finally, we would like to thank Asiya Guasa and Rodney Colón-Reyes for proofreading this manuscript.

**Conflicts of Interest:** The authors declare no conflict of interest.

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
