# Peer review of "Cryptococcus gattii Can Use the Cactus Pilosocereus spp. to Grow and Develop a Capsule and Produce Melanin In Vitro"

_2036-7481, doi:10.3390/microbiolres14020056_

Round 1
Reviewer 1 Report
Comment : The title needs revision. In its current form is confusing to the reader.
Comment: Abstract Include a small conclusion at the end of the abstract
Comment: In the introduction, you must connect the state of the art to the goals of your manuscript. Following the literature review, please provide a clear and concise interpretation of the present-day state of the art. It should clearly demonstrate the identified knowledge gaps and link them to the paper's objectives. Please justify both the novelty and importance of your paper's objectives. Discuss the previous studies to which you are referring in detail.
Comment: clear the statistical analysis in the manuscript
comment what is the novelty of present work.
Comment: In general, there are typo errors and grammar mistakes. Please, make sure about them.
comment Most of methods are needed to add references.
Comment: Please, do add more recent articles to discuss, and explain your results.
line 169 writ all genera C. gattii in italic name
How conclusion long than abstract
references not in journal stile
need to edit
Reviewer 2 Report
Ramos-Irizarry et al. investigate the ability of C. gattii to grow on different media containing different part of the cactus Pilosocereus spp. The results are interesting since they report new insights on a new ecological niche of this pathogen.
Here below authors can find my comments that could improve the manuscript.
General comments
- Manuscript needs a more careful revision of English that presents some mistakes and often the sentences are not very clear.
- There are many editing mistakes that need to be corrected along all the text: 1) The name of the species should be written in italic; 2) the generic species should be indicated with “spp.” (not italic); 3) once an abbreviation has been specified it should be used in all the text; 4) “inner part medium”, “outer part medium”, and “whole cactus medium” should be written all lowercase; 5) use in the correct way the words “cactus” (singular) or “cacti” (plural), ex. “cactus media” not “cacti media”. 6) when it is at the beginning of a sentence, after a full stop, write Cryptococcus gattii in full extension.
Title
It presents some mistakes. The right form is the following: Cryptococcus gattii can use different parts of the cactus Pilosocereus spp. to grow and produce melanin
Abstract
Line 11. Delete “(C. gattii)” since it is not necessary.
Line 13. Use “However” instead of “Although”
Line 15. “Cactus media” not “cacti media”
Line 16. “from” should not be in italic
Line 16. Delete “2.5x105” and write Cryptococcus in full extension.
Line 18. Same as line 15.
Line 20. Change “..OPM may have a precursor..” to “…outer part medium may contain a precursor..”
Line 22. Use “cactus” not “cacti”.
Introduction
The different molecular types/species of C. gattii are never introduced. Please add a sentence about this issue.
Lines 26-27. Only C. gattii VGI and VGII are able to infect also immunocompetent hosts, VGIII and VGIV infects immunocompromised hosts. Please modify the sentence.
Lines 30-31. This is not completely true. VGI and VGII in immunocompetent patients usually imply pulmonary manifestations without brain involvement. Modify the sentence.
Line 33. Why “hyperendemic”?
Line 45-46. There are not any phylogenetic evidences of dispersion of C. gattii from Australia and in case it is not surely a recent event.
Line 68-71. The C. gattii isolates from Northern Europe are all from clinical cases probably imported. The only environmental isolation of Cryptococcus gattii was from a Douglas fir in the Netherlands (Chowdhary et al. 2012). All the other environmental isolates in Europe are from southern countries.
Line 72. Mediterranean climate is characterized by dry and hot summers and mild winters.
Lines 84-85. Cryptococcus gattii was recovered only from one cactus Pilosocereus (Cephalocereus) royenii.
Materials and methods
Line 114. Abbreviation for potato dextrose agar should be PDA and for broth PDB.
Line 115. It is not clear why PDA is similar to cactus nutrient environment
Line 117. Add that R265 belong to VGII molecular type as the strain isolated in Puerto Rico in the cactus.
Line 132. Delete brackets for Dragotakes and Casadevall 2018.
Results
In the text, report p values as p<0.001 or p<0.05 without asterisks.
Paragraph 3.3. Results concerning WCM are not reported. It is important to know if on that medium the C. gattii isolate produced melanin or not.
Discussion
- Remember you have tested only one VGII strain and therefore your results could be different if tested using strains belonging to other molecular types/species.
- Along all discussion, description of results are repeated, please keep only comments and delete what was already reported in the previous section.
Line 287-288. Again the only cactus positive was P. royenii.
Line 300. In the present research any of the experiments was addressed to test the fertility of the strain. Delete “reproduce”.
Line 3161-362. Why are you sure that L-dopa is not present in the cactus media?
English language used in the manuscript presents some mistakes.
Round 2
Reviewer 1 Report
Very thanks for responed to reviewers
Cryptococcus gattii must be changed to C. gattii except first mention
Minor editing of English language required